# Burden of invasive group B Streptococcus disease in non-pregnant adults: A systematic review and meta-analysis

**Adoración Navarro-Torné[1], Daniel Curcio [1]\*, Jennifer C. Moïsi[2], Luis Jodar[3]**

**1** Pfizer Spain, Madrid, Spain, **2** Pfizer France, Paris, France, **3** Pfizer Inc, Collegeville, PA, United States of America

\* Daniel.Curcio@pfizer.com

## Abstract

### Background

*Streptococcus agalactiae* or group B Streptococcus (GBS) has emerged as an important cause of invasive disease in adults, particularly among the elderly and those with underlying comorbidities. Traditionally, it was recognised as an opportunistic pathogen colonising and causing disease in pregnant women, neonates, and young infants. Reasons for the upsurge of invasive GBS (iGBS) among the elderly remain unclear, although it has been related to risk factors such as underlying chronic diseases, immunosenescence, impaired inflammatory response, and spread of virulent clones. Antibiotics are successfully as treatment or prophylaxis against iGBS. Several candidate vaccines against iGBS are under development.

### Objectives

To conduct a systematic review of the current literature on invasive GBS in order to determine disease incidence and case fatality ratio (CFR) among non-pregnant adults. Additionally, information on risk factors, clinical presentation, serotype distribution, and antimicrobial resistance was also retrieved.

### Methods

Between January and June 2020, electronic searches were conducted in relevant databases: MEDLINE, EMBASE, Global Health, and SCOPUS. Studies were included in the systematic review if they met the inclusion/exclusion criteria. The authors assessed the selected studies for relevance, risk of bias, outcome measures, and heterogeneity. Meta-analyses on incidence and CFR were conducted after evaluating the quality of methods for assessment of exposure and outcomes.

### Results

Pooled estimates of iGBS incidence in non-pregnant adults 15 years and older were 2.86 cases per 100.000 population (95% CI, 1.68–4.34). Incidence rates in older adults were substantially higher, 9.13 (95%CI, 3.53–17.22) and 19.40 (95%CI, 16.26–22.81) per 100.000

**Data Availability Statement:** All relevant data are within the manuscript and its Supporting Information files.

**Funding:** The authors did not receive financial support for the conduct of this research apart from their salary as full-time employees of Pfizer. The funders had no role in study design, data collection and analysis, decision to publish, or preparation of the manuscript.

**Competing interests:** All authors are employees of Pfizer and may hold stock options. Pfizer supported the work.

population ≥50 and ≥ 65 years old, respectively. Incidence rates ranged from 0.40 (95% CI, 0.30–0.60) in Africa to 5.90 cases per 100.000 population (95% CI, 4.30–7.70) in North America. The overall CFR was and 9.98% (95% CI, 8.47–11.58). CFR was highest in Africa at 22.09% (95% CI, 12.31–33.57). Serotype V was the most prevalent serotype globally and in North America accounting for 43.48% (n = 12926) and 46,72% (n = 12184) of cases, respectively. Serotype Ia was the second and serotype III was more prevalent in Europe (25.0%) and Asia (29.5%). Comorbidities were frequent among non-pregnant adult iGBS cases. Antimicrobial resistance against different antibiotics (i.e., penicillin, erythromycin) is increasing over time.

## Conclusions

This systematic review revealed that iGBS in non-pregnant adults has risen in the last few years and has become a serious public health threat especially in older adults with underlying conditions. Given the current serotype distribution, vaccines including serotypes predominant among non-pregnant adults (i.e., serotypes V, Ia, II, and III) in their formulation are needed to provide breadth of protection. Continued surveillance monitoring potential changes in serotype distribution and antimicrobial resistance patterns are warranted to inform public health interventions.

## Introduction

*Streptococcus agalactiae* or Group B *Streptococcus* (GBS) is a Gram-positive microorganism with a polysaccharide capsule characterised by the cell-wall-specific Lancefield´s Group B antigen [1]. The capsular polysaccharide is a principal virulence factor and is associated with invasive capacity. Moreover, different serotypes vary in invasive potential [2] and their distribution varies by age group and geographic region [3]. GBS is part of the normal gastrointestinal and genitourinary flora of healthy adults and acts as an opportunistic pathogen, developing from asymptomatic carriage to non-invasive or invasive disease [4]. GBS causes a range of maternal-foetal illnesses during pregnancy and post-partum, from mild urinary tract infections to chorioamnionitis and sepsis in pregnant women to severe neonatal invasive disease such as meningitis or sepsis [5], which may lead to severe impairment or death. GBS colonization in pregnancy has also been associated with an increased risk of prematurity and stillbirth.

GBS infections in non-pregnant adults, particularly among the elderly, have emerged as an important pathogen in this age group [6]. Invasive GBS (iGBS) disease is a major clinical entity: the most common presentation is primary bacteraemia [6], followed by skin and soft tissue infection [7], pneumonia, urosepsis, endocarditis, peritonitis, meningitis, and empyema [8]. An increase in the incidence of iGBS in adults over time has been observed [6, 9] and relapse is relatively frequent [7]. In particular, older age (i.e., ≥65 years) has been associated to increasing iGBS disease incidence and mortality, and 50% of lethal GBS infections occur in the elderly [6]. Most of the cases in older adults are linked to underlying medical conditions such as diabetes mellitus, obesity, liver cirrhosis, stroke, cancer, and cardiovascular disease [10, 11] and with immunosenescence [12]. Intrapartum antibiotic prophylaxis and antibiotic treatment are successfully used to prevent and treat GBS infections. However, recent reports have described increasing antimicrobial resistance [2, 13, 14], highlighting the need for a universal GBS vaccine that helps protect not only infants and pregnant women but also older adults with underlying comorbidities.

We conducted a systematic literature review and meta-analysis of the incidence of iGBS disease and the case-fatality ratio of iGBS among non-pregnant adults worldwide. We also described the risk factors, serotype distribution, and antimicrobial resistance patterns of iGBS disease among non-pregnant adults.

## Methods

### Data searches

We identified data through systematic review of the published literature including Medline (via OVID), Embase (via OVID), Global Health (via OVID), and Scopus from inception through June 2020. We searched databases with variants of terms "streptococcus", "streptococcal infections", "*Streptococcus agalactiae*", "agalactiae" or "group b", "invasive", and "virulent". Medical subject heading (MeSH) terms were used where possible. The full search strategy is shown in Fig 1 [15]. One investigator (A.N.T.) performed the databases searches, two independent reviewers (A.N.T. and D.C.) screened the titles for duplicates and for eligibility, and screened abstracts to assess the suitability for inclusion, and one reviewer (A.N.T.) extracted the data.

### Exposure

Case definition for iGBS includes the laboratory isolation of GBS (*Streptococcus agalactiae*) or detection of GBS antigen or its nucleic acid in any normally sterile site (i.e., blood, deep tissue, CSF, joint fluid, pleural fluid, etc.).

### Outcomes

The outcomes of interest were the incidence and case fatality ratio due to iGBS. If the incidence of iGBS was not reported, the paper was included if it provided information on the

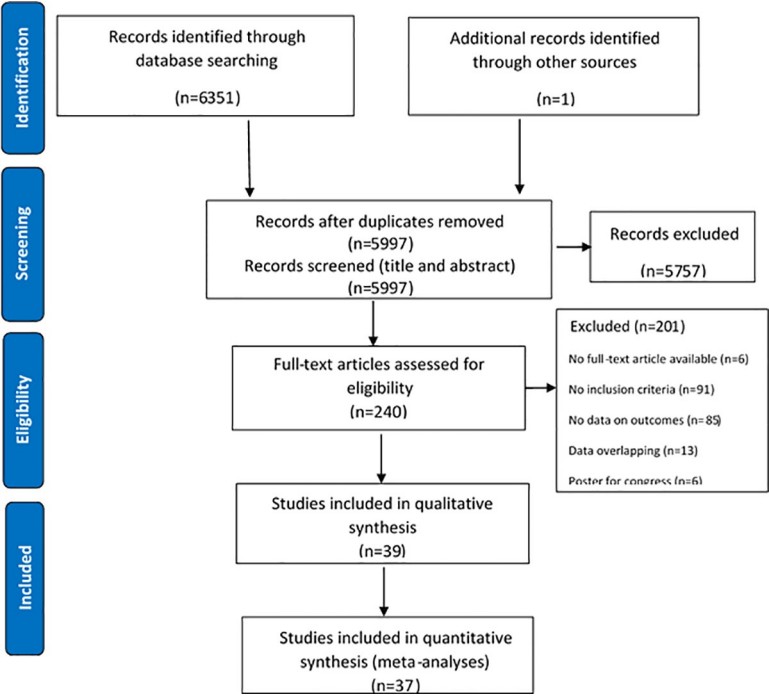

**Fig 1. PRISMA flow diagram of selection of studies.**

number of participants with iGBS (number of cases) and the population denominator (i.e., catchment population of the hospital, population covered by surveillance). Incidence was calculated as the number of iGBS cases divided by population at risk during the study period, and it was expressed as a rate per 100.000 population per year. Articles that expressed incidence in other forms (i.e., incidence per hospital admissions or discharges) were not considered for the meta-analysis on incidence but were retained if they provided information on case-fatality ratio (CFR). When denominator data were not explicitly provided, denominators were extrapolated using data describing the total number of cases of iGBS infection and the incidence rate. In some instances, incidence was given for the first and last year of the study period, thus final incidence was calculated as the average between these two values. In other cases, incidence was given for two or more adult age groups and overall incidence was calculated as the average of the values of the different age groups.

In articles where CFR was not explicitly reported, CFR was calculated using available data (i.e., by dividing the number of deaths attributable to iGBS infection by the number of iGBS cases).

Time periods of 15 years were considered to examine temporal patterns for each of the outcomes of interest.

## Inclusion and exclusion criteria

We included all observational studies that referred to iGBS in non-pregnant adults. If studies covered other age groups, the study was included but only data concerning non-pregnant adults ($\geq$ 15 years old) was considered. Only journal articles that included an abstract were considered. There were no time or geographic restrictions and languages were restricted to English, Spanish, French, Portuguese, and Italian. Infections other than iGBS were not included (i.e., urine, wound). However, papers describing skin/soft tissue infections were included if the authors of the paper considered these as invasive disease. We assumed the terminology "invasive" was clinically justified, i.e., that the skin and soft tissue infections were confirmed by isolation of GBS from deep tissue or associated with concomitant bacteraemia.

## Data collection

The authors selected the studies in the electronic databases according to the above-mentioned search strategies and transferred them into Endnote X9 Reference Manager Software®. Duplicates were removed and the remaining studies were scrutinised for relevance. Titles and abstracts were reviewed in detail and the eligible studies for iGBS were retained. After discarding irrelevant papers, full-text articles were reviewed in depth and papers that met the inclusion/exclusion criteria and case definition were considered for this review. Even if relevant for this review, papers were excluded if data overlapped with other studies (i.e., covered the same geographical area and/or the same time period).

Data on study characteristics and results were extracted to a pre-set Excel abstraction form.

## Assessment of the quality

The quality of the articles was assessed using the Joanna Briggs Institute´s Critical Appraisal Tool for prevalence/incidence systematic reviews [16, 17]. This Critical Appraisal Tool provides a checklist that covers nine domains: appropriateness of sample frame, recruitment of participants, adequacy of sample size, description of study subjects and setting, coverage of identified samples, valid methods for identification of the condition, standardized and reliable measurement of the condition, appropriateness of the statistical analysis, and adequacy of the response rate.

### Data analysis

Only papers that included incidence of iGBS or CFR were considered. The authors decided to pursue a pooled analysis of these papers and a quantitative synthesis and meta-analyses were undertaken.

Random-effects meta-analyses were conducted with STATA 14 software (StataCorp®) to better address heterogeneity when estimating the incidence of iGBS and case fatality ratio in non-pregnant adults. To assess the impact of several covariates (region, time period, diabetes, and cardiovascular disease) on incidence and CFR estimates from the meta-analyses, a meta-regression method was utilised. A subgroup analysis of incidence stratifying by age was performed to assess the impact of ageing on the iGBS incidence.

## Results

### Characteristics of included studies

The search strategy identified 6351 references from the selected electronic databases (Fig 1). Hand searches identified one additional paper. After removing duplicates and screening titles and abstracts, 240 full text articles were reviewed in detail. Of those, 39 articles met the inclusion criteria for this review (Table 1). [2, 18–55], 25 of which were included in a meta-analysis of the incidence of iGBS and 31 in the meta-analysis of the CFR.

Out of the 29 articles that provided data on incidence, all but five [21, 26, 35, 42, 44] reported these data in the appropriate format (cases/100,000) to be included for the quantitative analysis. Eighteen articles (Table 1) reported population-based surveillance studies, some of them nationwide surveillance, whereas eight were prospective studies and fourteen reported on retrospective studies (Table 1). Twenty-three articles were hospital-based and fourteen laboratory-based surveillance. All papers but six provided a case definition for iGBS [20, 24, 26, 28, 35, 43]. Of the selected articles, thirteen articles collected data from Europe, thirteen papers reported on data from North America, two from South America, two from Africa (the paper by Camuset *et al.* [24] from the Réunion Island was considered geographically ascribed to Africa although administratively belongs to France), eight from Asia, and one from Australia.

Of the 39 articles to be included either in the qualitative or the quantitative synthesis, all of them (100%) were considered to respond affirmatively to the question "Was the sample frame appropriate to address the target population?". The same proportion was applicable to all the selected papers in response to the questions "Were study participants sampled in an appropriate way?", "Was the sample size adequate?", and "Was the data analysis conducted with sufficient coverage of the identified sample?". To the question "Were the study subjects and the setting described in detail?", 89.7% of the papers were assessed as positively whereas in 12.8% of the papers it was considered unclear. In 94.9% of the selected papers, the use of valid methods for the identification of the condition was established while in 5.1% of the papers it was unclear. The same percentages apply to the assessment of whether the condition was measured in a standard, reliable way for all participants. In relation to the appropriateness of the statistical analysis, it was not applicable to 20.1% of the papers. In conclusion, all the selected papers were included for the analyses after the application of the Joanna Briggs Institute´s Critical Appraisal Tool as their quality was deemed adequate (S4 Table).

Overall, between-study heterogeneity was very high (>90%) in most of the study sub-groups.

### Incidence of invasive group B streptococcal infection

There were 66292 non-pregnant adults with invasive GBS in a population at risk of 1.534.695.818 individuals across 13 countries considered for the quantitative analysis. The

**Table 1. Summary of included studies.**

| Author | Study period | Region | Study design | Age range (years) | No of iGBS cases | Incidence (cases per 100.000) | Method for incidence estimation | CFR (%) |
|---|---|---|---|---|---|---|---|---|
| Alhhazmi | 2003–2013 | North America | Population-based surveillance | ≥ 15 | 1372 | 3.23 | Average of incidence by age group provided by article | - |
| Barnham | 1978–1988 | Europe | Population-based surveillance | 25–86 | 6 | 0.24 | Estimated from cases per catchment population provided by article | 33.0 |
| Bjornsdottir | 1975–2014 | Europe | Population-based surveillance | 19–90 | 139 | 2.19 | Average of incidence by age group provided by article | - |
| Blumberg | 1992–1993 | North America | Population-based surveillance | 39.1–75.3 | 112 | 5.9 | Provided by article | - |
| Bolaños | 1992–1999 | Europe | Population-based surveillance | 21–100 | 32 | 1.5 | Provided by article | 31.0 |
| Bunyasontigul | 1999–2009 | Asia | Retrospective cohort study | 15.3–91.6 | 101 | - | - | 7.9 |
| Camuset | 2011 | Africa | Prospective hospital-based study | 25–93 | 22 | 10.1 | Provided by article | 4.5 |
| Collin | 2015–2016 | Europe | Population-based surveillance | ≥ 15 | 2225 (episodes) | 2.9 | Provided by article | 12.5 |
| Cooper | 1991–1996 | North America | Retrospective hospital-based study | "adults" | 55 | - | - | 16.4 |
| Crespo-Ortiz | 2004–2012 | South America | Retrospective & cross-sectional study | 17–83 | 57 | 0.90 | Estimated from cases per catchment population provided by article | 17.5 |
| Darbar | 2000–2005 | Australia | Prospective hospital-based study | 47.3–78.7 | 80 | - | - | 10.0 |
| Farley | 1989–1990 | North America | Population-based surveillance | 18–99 | 140 | 4.4 | Provided by article | 21.0 |
| Francois Watkins | 2008–2016 | North America | Population-based surveillance | ≥ 18 | 21250 | 9.5 | Average of incidence of surveillance years provided by article | 6.5 |
| Fujiya | 2002–2014 | Asia | Retrospective cohort study | 24–91 | 52 | 2.5 | Provided by article | 5.8 |
| Georges | 1991–2006 | Europe | Population-based surveillance | 15–64 | - | 1.8 | Provided by article. Not used for the meta-analysis due to information on denominators not available | - |
| | | | | >64 | | 4.1(1991) 9.1 (2006) | | |
| Gimenez | 1983–1993 | Europe | Retrospective study | 40–80 | 35 | 0.58 | Estimated from cases per catchment population provided by article | 8.7 |
| Gudjonsdottir | 2004–2009 | Europe | Population-based surveillance | 23–103 | 317 | 3.47 | Estimated from cases per catchment population provided by article | 12.0 |
| Huang | 2001–2003 | Asia | Retrospective study | 22–89 | 94 | - | - | 20.2 |
| Jenkins | 2006–2009 | Europe | Population-based surveillance | 28–84 | 17 | 0.69 | Provided by article | 0.0 |
| Jones | 2000–2003 | Europe | Prospective hospital-based study | > 60 | 70 | 11.0 | Estimated from cases per catchment population provided by article | - |
| Jump | 2008–2017 | North America | Prospective cohort study | ≥ 18 | 5497 | - | - | |
| Kalimuddin | 2011–2015 | Asia | Retrospective cohort study | Nonpregnant adults | 408 | - | - | 7.4 |
| Lamagni | 1991–2010 | Europe | Retrospective study | ≥ 15 | 13376 | 1.66 | Average of incidence of surveillance years provided by article | - |
| Lambertsen | 1999–2004 | Europe | Population-based surveillance | 16–99 | 411 | 2.7 | Average of incidence of surveillance years provided by article | 14.0 |
| Lee | 1991–1999 | Asia | Prospective hospital-based study | 18–85 | 71 | - | - | 7.0 |
| Lopardo | 1998–1999 | South America | Prospective hospital-based study | 21–83 | 31 | - | - | 12.9 |
| Matsubara | 1998–2007 | Asia | Retrospective hospital-based study | 29–90 | 52 | 1.04 | Estimated from cases per catchment population provided by article | 13.5 |

*(Continued)*

**Table 1.** (Continued)

| Author | Study period | Region | Study design | Age range (years) | No of iGBS cases | Incidence (cases per 100.000) | Method for incidence estimation | CFR (%) |
|---|---|---|---|---|---|---|---|---|
| Morozumi | 2010–2013 | Asia | Prospective study | 19- >90 | 443 | - | - | 10.2 |
| Mosites | 2002–2015 | North America | Retrospective study | ≥ 18 | 6 | - | - | 33.0 |
| Perovic | 1995–1997 | Africa | Retrospective hospital-based study | 22–87 | 40 | 0.44 | Estimated from cases per catchment population provided by article | 35.0 |
| Phares | 1999–2005 | North America | Population-based surveillance | ≥ 15 | 11663 | 6.95 | Average of incidence of surveillance years provided by article | 10.3 |
| Ruppen | 1998–2015 | Europe | Retrospective hospital-based study | ≥ 65 | 171 | - | - | 5.0 |
| Schrag | 1993–1998 | North America | Population-based surveillance | ≥ 15 | 5293 | 11.9$^\alpha$ | Average of incidence by age group provided by article | 11.5 |
| Schwartz | 1982–1983 | North America | Retrospective population-based surveillance | ≥ 20 | 56 | 2.4 | Provided by article | - |
| Shelburne | 2000–2011 | North America | Retrospective hospital-based study | 27–86 | 147 | - | - | 6.0 |
| Skoff | 2007 | North America | Population-based surveillance | 18–105 | 1546 | 7.3 | Provided by article | 7.5 |
| Slotved | 2005–2018 | Europe | Population-based surveillance | 20–64 | - | 1.35 | Not used for the meta-analysis. Number of cases by age group not extractable | - |
| | | | | 65–74 | | 5.35 | | |
| | | | | > 75 | | 9.80 | | |
| Tyrrell | 1996 | North America | Population-based surveillance | ≥ 15 | 91 | 4.1 | Provided by article | 5.5 |
| Wilder-Smith | 1998 | Asia | Prospective hospital-based study | 24–63 | 11 | 1.1 | Estimated from cases per catchment population provided by article | 9.1 |

α: Incidence in 2008 only.

incidence rate for iGBS among non-pregnant adults was 2.86 cases per 100.000 population (95% CI, 1.68–4.34) (Fig 2) overall, and was 5.90 cases per 100.000 population (95% CI, 4.30–7.70) in North America, 1.50 (95% CI, 1.10–2.00) in Europe, 1.50 (95% CI, 0.70–2.60) in Asia, 0.90 (95% CI, 0.70–1.20) in South America (although there was only one article from this region), and 0.40 (95% CI, 0.30–0.60) in Africa.

The literature review covered a period from 1975 to 2018. There were 237 iGBS cases in sub-period 1 (1975–1990), 39197 iGBS cases in sub-period 2 (1991–2005), and 26858 iGBS cases in sub-period 3 (2006–2018). Overall incidence increased from 1.50 (95% CI, 0.26–3.75) in sub-period 1975–1990 to 2.73 (95% CI, 1.05–5.19) in sub-period 1991–2005 and 3.79 per 100.000 population (95% CI, 1.90–6.34) in sub-period 2006–2018 (Fig 3).

The meta-regression showed no significant effect of the region or the study period on disease incidence (p = 0.993, and p = 0.234, respectively).

A sub-analysis of incidence data stratified by age showed a pooled estimate for incidence of 9.13 per 100.000 population (95%CI, 3.53–17.22) in adults of 50 years of age or over [18, 24, 25, 27, 37, 47, 49, 54] and 19.40 per 100.000 population (95%CI, 16.26–22.81) in 65 years or over [24, 47, 49, 54].

## Case fatality risk of invasive group B streptococcal infection

There were 4379 deaths attributable to iGBS among 49867 non-pregnant adult cases of iGBS among studies considered for the quantitative analysis. The overall CFR was 9.98% (95% CI,

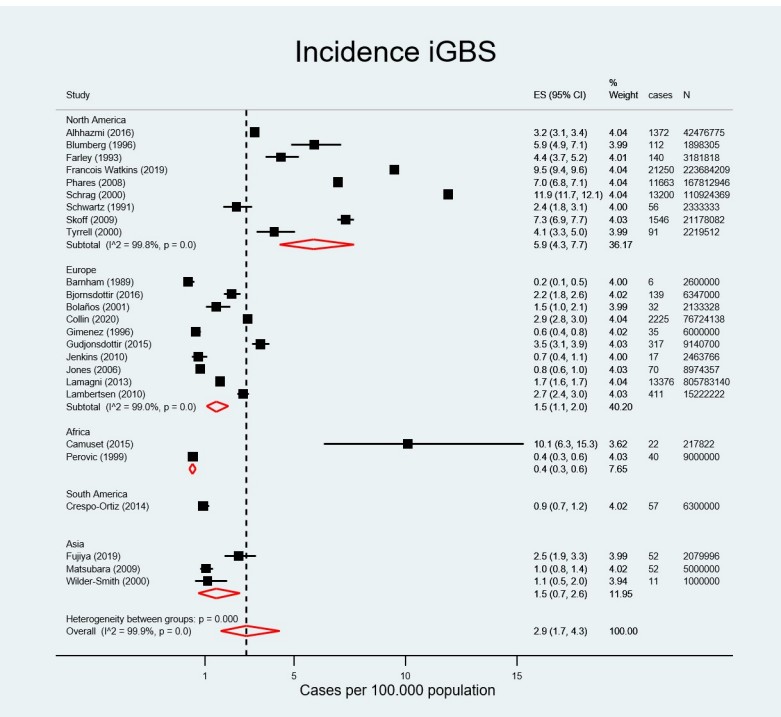

**Fig 2. Pooled estimated incidence of invasive group B streptococcal infection by region.**

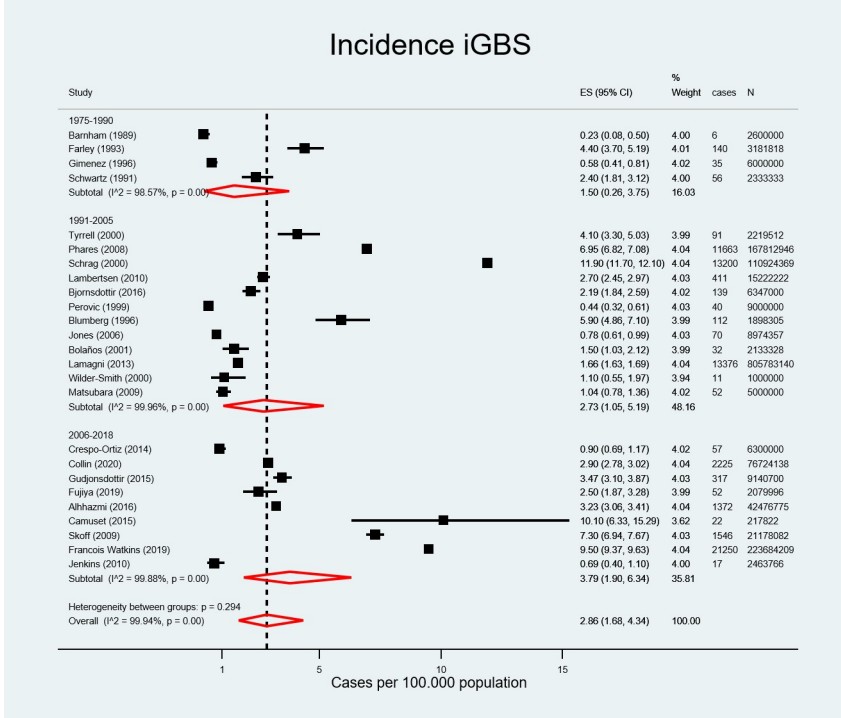

**Fig 3. Pooled estimated incidence of invasive group B streptococcal infection by period.**

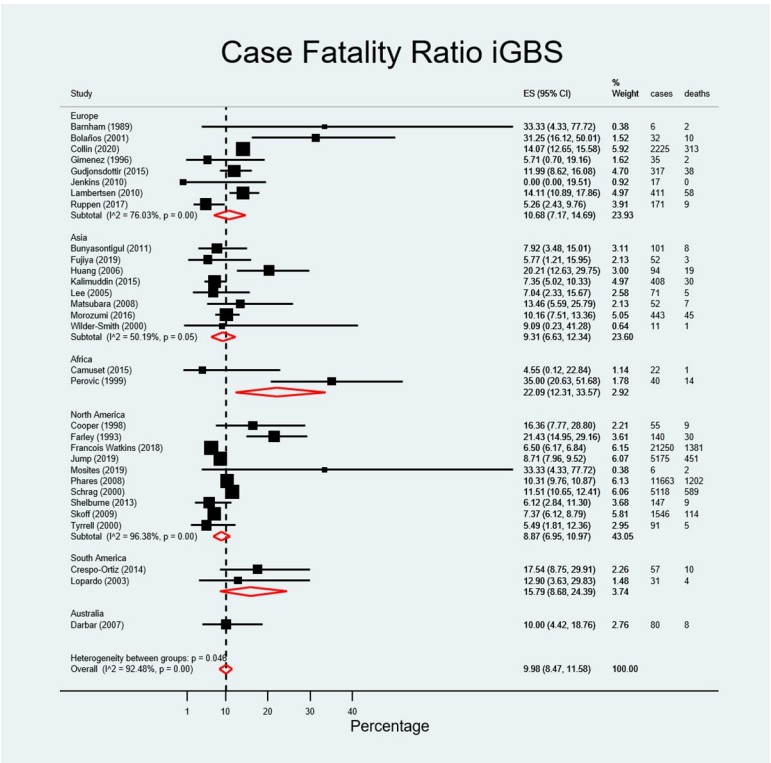

**Fig 4. Pooled estimated case fatality ratio of invasive group B streptococcal infection by region.**

8.47–11.58): it was 9.31% (95% CI, 6.63–12.34) in Asia, 8.87% (95% CI; 6.95–10.97) in North America, 10.00% (95%CI, 4.42–18.76) in Australia (one paper), 10.68% (95% CI, 7.17–14.69) in Europe, and 15.79% (95%CI, 8.68–24.39) in South America. CFR in Africa was 22.09% (95% CI, 12.31–33.57) although there were only two reports from this region (Fig 4). The meta-regression showed no significant effect of the region on the CFR (p = 0.758). CFR decreased over time, from 15.12% (95% CI, 3.37–31.67; 34 deaths among 181 cases) in 1975–1990, to 11.83% (95%CI, 9.96–13.84; 1939 deaths among 17850 cases) in 1991 to 2005, to 7.91% (95% CI, 6.11–9.90; 2406 deaths among 31836 cases) in 2006 to 2017 (Fig 5). This effect was borderline statistically significant (p = 0.05 in meta-regression).

### Secondary outcomes

**Risk factors for invasive group B streptococcal infection.** In 26 out of the 39 selected studies [2, 22–28, 30, 33–36, 39, 41–43, 46–52, 54, 55], diabetes was identified as the main underlying co-morbid condition ranging from 15 to 64% of the iGBS cases [22, 24, 41]. Cancer and malignancies (including haematological malignancies) were also a frequent comorbidity of iGBS patients [2, 22, 24–28, 30, 33–35, 41–43, 47, 49–52, 55] ranging from 7 to 25%, and also to a lesser extent cardiovascular disease, high blood pressure, liver cirrhosis, renal disease, obesity, alcohol abuse, neurologic disorders, lung disease, treatment with corticosteroids [42, 55] and hospitalisation or surgical procedures [22, 46, 52].

The meta-regression of diabetes and cardiovascular disease covariates on the incidence of iGBS showed a trend of a slight increase in incidence with increasing prevalence of the covariates, although this association was not statistically significant.

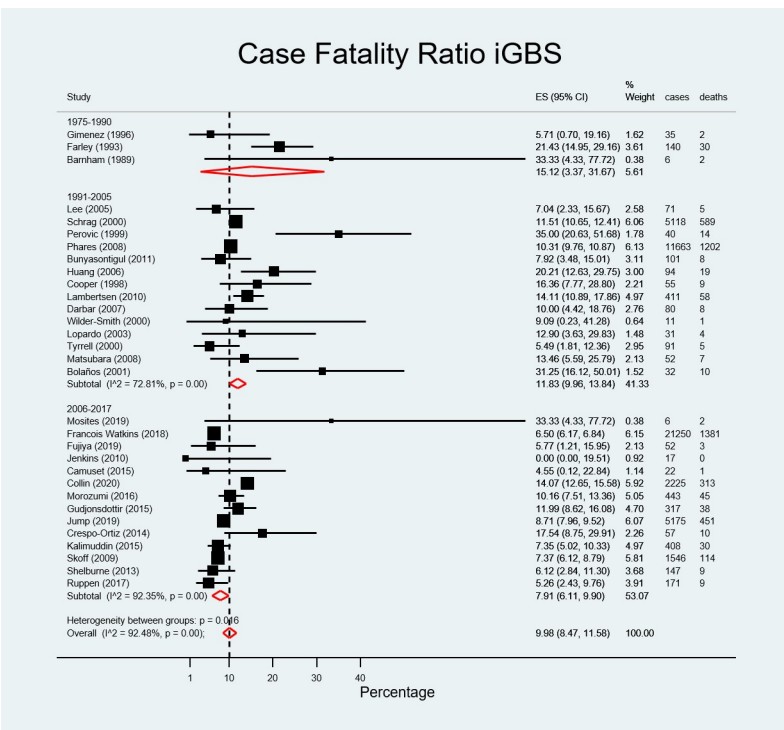

**Fig 5. Pooled estimated case fatality ratio of invasive group B streptococcal infection by study period.**

**Serotype distribution.** A total of 29731 isolates from non-pregnant adults were serotyped of which serotype V was the most common accounting for 43.48% (n = 12926) of isolates, followed by serotype Ia, 18.31% (n = 5443) and serotype III, 11.72% (n = 3483). In relation to serotype distribution by region, serotype V predominated in North America whereas serotype III was more prevalent in Europe and Asia. In South America, there were no isolates of serotype V. However, data from Africa (n = 17), Australia (n = 66), and South America (n = 31) should be taken cautiously due to the low number of isolates in these regions. In Asia, serotype VI represented 11.31% of the serotyped isolates, in contrast with other regions where prevalence of this serotype is very low or inexistent.

**Antimicrobial resistance.** Nineteen studies reported information on antimicrobial susceptibility testing. In fifteen of them, all tested isolates were 100% susceptible to penicillin [2, 22, 24, 28, 31, 36, 40, 42–44, 46, 47, 51, 54, 55]. Four studies reported some resistance to penicillin [23, 27, 30, 35], at 2%, 1.4%, 0.5%, and 2%, respectively (S5 Table). Resistance to erythromycin was reported in fourteen of the nineteen studies with data on antimicrobial susceptibility testing, ranging from 2% [44] to 54.8% of isolates [30]. Resistance was also reported for tetracycline with 83.9%, 95%, 72.4%, and 89% of the tested isolates resistant to tetracycline, respectively [30, 42, 43, 51]. Of note, antimicrobial resistance increased over time, both in number of antibiotics and in proportion of isolates, based on studies that covered large periods of time [30]. For instance, resistance to clindamycin increased over time, reported by Francois Watkins *et al.* [30] at 43.2% during the period 2008–2016 whereas Bolaños *et al.* [22] described all their isolates as susceptible in 1992–1999. Alarmingly, resistance to penicillin reached high levels (i.e., minimum inhibitory concentration, MIC > 8 μg/mL) in some settings, as reported by Crespo-Ortiz in the last period of their surveillance, 2012 [27].

## Discussion

This systematic review and meta-analysis presents data on invasive GBS cases from non-pregnant adults across the world between 1975 and 2018. To our knowledge, this is the first systematic review and meta-analysis that addresses the global burden of iGBS disease among non-pregnant adults 15 years and older.

The pooled estimated incidence of iGBS in non-pregnant adults was 2.86 cases per 100.000 population (95% CI, 1.68–4.34). This meta-analysis included several studies from nationwide population-based surveillance systems [25, 30, 40, 47, 49, 52] with a large number of cases and populations at risk originating mainly from the USA, Canada, and Europe. In contrast, studies from Africa and South America were narrower in scope, which may have contributed to the regional disparities in incidence rates. Methodological differences in case ascertainment and detection, ethnic differences [30] and higher prevalence of underlying conditions such as diabetes among adult populations in industrialized regions may also explain the increased iGBS rates in these regions compared to other regions [56].

Incidence of iGBS is higher in adults [9, 25, 40, 58], and particularly in older adults compared to neonates and infants and appears to be increasing over time [30, 47]. The stratified analysis of the iGBS incidence showed a threefold increase in incidence from 2.86 cases per 100.000 population (95% CI, 1.68–4.34) among the overall adult population to 9.13 cases per 100.000 population (95%CI, 3.53–17.22) in adults ≥ 50 years. Incidence in adults ≥ 65 years was even higher, at 19.40 per 100.000 population (95%CI 16.26–22.81), suggesting that iGBS incidence increases with age. Factors contributing to the rising incidence of iGBS among older adults, are higher prevalence of underlying chronic diseases, particularly diabetes, obesity, malignancies, cardiovascular disease, liver and renal disease, and alcoholism, among others [2, 22, 25–28, 30, 33–35, 41–43, 47, 49–52, 55] and immunosenescence or ageing altered cell-mediated immunity [35, 57].

An additional reason for an iGBS increased incidence in North America and Europe might be that a large number of elderly people in these countries live in long-term care or skilled nursing facilities, where invasive medical devices such as urinary or intravenous catheters are extensively used and person-to-person transmission may occur [58].

Important differences in iGBS incidence, even between countries from the same geographic subgroup and time period, were observed. Several explanations for these disparities are plausible. Differing medical practices may have had an impact, with some physicians requesting a microbiological investigation of the suspected cases whereas others may have preferred an empirical treatment without further microbiological information. Moreover, physicians may have participated in a specific training on iGBS for surveillance purposes as denoted by Schrag et al. [49], increasing their awareness and therefore, the potential detection of iGBS. In addition, laboratory practices may also differ, as not all laboratories are equally accurate and qualified for the isolation and characterisation of GBS. Socioeconomic factors and ethnicity may have affected the surveillance, as described by Schrag et al. [49]. In their study, they determined that the risk of invasive disease among black adults was twice than among white adults and this difference was persistent over time. Hence, different population composition may lead to dramatic differences in incidence due to the ethnicity or socioeconomic gap. Different age distributions may also play a role in incidence disparities. Underlying conditions may predispose to GBS infection, particularly among the elderly. Studies considering a more numerous elderly age group may present a higher incidence [54] due to older age but also to increased comorbidities.

Mortality in this review was determined at 9.98% (95%CI, 8.47–11.58) overall, ranging from 9.31% (95% CI, 6.63–12.34) in Asia to 22.09% (95% CI; 12.31–33.57) in Africa. However,

these figures should be interpreted prudently since some of these studies comprised low numbers of cases and deaths. Of note, when CFR was broken down by study period it showed a decline from 15.12% (95% CI, 3.37–31.67) in the sub-period 1975–1990 to 7.91% (95% CI, 6.11–9.90) in 2006–2017, probably reflecting an improvement in healthcare facilities, treatment options and services over time. The meta-regression analysis confirmed this decline to be significant. Beyond socio-economic conditions and robustness of healthcare facilities, disparities in CFR across regions may reflect different clinical practices, such as use of blood cultures for diagnosis but also delayed diagnosis in the elderly due to an impaired inflammatory response or masked physical signs which may lead to fatal outcomes [59].

Serotype distribution varies across regions and age groups. In contrast to infants and neonates where serotype III tends to predominate [28, 34], serotype V was the major contributor to the rise of iGBS incidence in non-pregnant adults [60] overall, perhaps due to the acquisition of genetic determinants of antimicrobial resistance [12, 47, 58, 60–62], particularly, erythromycin resistance [47]. Of note, some researchers have reported higher CFR for serotype V compared to other serotypes [28]. As with incidence rates and CFR, serotype distribution is also different per region. For example, whereas serotype V was predominant in North America, serotype III was more prevalent in Europe and Asia. The population-based surveillance of iGBS spanning from 2008 to 2016 in the USA highlighted the notable emergence and increase of serotype IV among non-pregnant adults between the start and the end of the study [30].

Most of the isolates reported in this review remained susceptible to penicillin. Recent studies in Portugal reported increases in macrolide and lincosamide resistance in GBS, even when consumption of macrolides decreased, suggesting that the successful expansion of certain clones was the major driver for this variation [13]. Likewise, later research from Japan [2] showed reduced penicillin susceptibility, and macrolide and quinolone resistance. In South America, consistent with global trends, findings show an increase in resistance to erythromycin and clindamycin, and the appearance of penicillin-non-susceptible strains [27]. Other reports echoed increases in clindamycin resistance [30] which may pose serious challenges to clinical management of penicillin-allergic patients. The presence of resistance to multiple antibiotics among iGBS adult disease is concerning. A GBS vaccine with broad serotype coverage may have value in reducing antimicrobial resistance, if it were implemented in key at risk populations [63].

In this review, iGBS has been found to be associated to diabetes and obesity, showing an increasing trend in multiannual studies [9]. Pitts *et al* estimated that the population-attributable risks of iGBS were 27.2% for obesity and 40.1% for diabetes [11]. Some articles report up to 10.5-fold higher risk of iGBS infection in persons with diabetes, and 16.4-fold higher in patients with cancer compared with the general population [50]. McLaughlin *et al* found that most GBS infections occurred in adults with chronic medical conditions where rates of GBS were 2 to 6 times higher compared with the general population [64]. Cardiovascular disease and alcohol-related liver damage are also important comorbidities in non-pregnant adults with iGBS.

Strengths of this review include the wide geographic scope, and the broad population-based studies, multiannual in many instances, which have allowed capturing of longitudinal changes and trends in the epidemiology of iGBS. The assessment of reports across the world has also permitted the evaluation of geographical differences and global challenges. The large number of overall iGBS cases in this review (n = 66,292) has allowed the calculation of pooled estimates with confidence. One limitation of our study is the heterogeneity observed in the meta-analyses. We applied a quality appraisal tool, and fitted random effects models and subgroup analyses to account for heterogeneity, but other methods may be appropriate as well [65, 66]. Notwithstanding heterogeneity, studies from different regions of the world reported on iGBS

incidence and CFR among persons similar in age and comorbidities, which may add to the generalisability of this review.

In considering this review, the potential underestimation of incidence rates should be borne in mind. For the estimation of individual incidence rates, in those papers not providing the overall incidence rate, the mean of the values given in the article for different adult age groups was calculated and this undoubtedly has decreased incidence rates since almost invariably, incidence rates were higher in elderly adults (i.e., 65 years or over), yet no weighting was performed. The same is true for incidence rates calculated as the average between the value of the starting and last year of the surveillance in multiannual studies. Difficulties in case ascertainment in low-, and middle-income countries may have contributed to lower incidence rates in Africa, Asia, and South America as well.

On the other hand, some studies were not powered to detect significant differences among subjects due to the small number of cases. Therefore, a call for prudence when interpreting these results is necessary. In contrast, large population-based surveillance studies may have driven this review.

## Conclusions

The findings of this review suggest that iGBS is a severe cause of non-pregnant adult disease, and the risk notably increases with age. In addition to increasing age, other risk factors for iGBS are diabetes, cancer, cardiovascular disease, liver disease, obesity, and other chronic conditions. CFR is also high, particularly in Africa and South America.

Antimicrobial resistance in iGBS is on the rise, with a considerable number of papers reporting on increases in resistance to several antibiotics. Alarmingly, increases in resistance to penicillin, the drug of choice for treatment and IAP of GBS, are now commonplace. A GBS adult vaccine would contribute to the reduction of overall and resistant infections. Moreover, improvement in resistant health outcomes may have economic ramifications at different levels [67].

GBS serotype distribution is also changing worldwide with the threat of emergence and spread of virulent lineages across the world. As revealed by this review, serotype distribution varies with age and across regions. Therefore, the development of an effective GBS vaccine should account for these differences and include serotypes that are predominant within the elderly population.

GBS prevention by targeted vaccination in adults holds promise although the potential for associated reductions in AMR and for protection against all manifestations of the disease, including non-invasive infections that have been found 3 to 4 times more frequent than invasive disease [64, 68], warrant further research.

## Supporting information

**S1 Checklist. PRISMA 2009 checklist.**
(DOC)

**S1 Fig. Serotype distribution by frequency.** NT = Nontypeable.
(TIF)

**S2 Fig. Serotype distribution by region.** NT = Nontypeable.
(TIF)

**S1 Table. Characteristics of the studies included in the systematic review and meta-analysis.**
(DOCX)

**S2 Table. Participant´s characteristics.** NA = Not available.
(DOCX)

**S3 Table. Outcomes of the studies included in the systematic review and meta-analysis.**
(DOCX)

**S4 Table. Application of the Johanna Briggs Institute Appraisal Tool for quality assessment.**
(DOCX)

**S5 Table. Reported antimicrobial resistance among the selected studies (%).** PEN Penicillin, ERY Erythromycin, GEN Gentamycin, S Susceptible, AMP Ampicillin, AZM Azithromycin, TCY Tetracycline, R Resistant, CEF Cephalothin, MIN Minocycline, CHL Chloramphenicol, I Intermediate, CTX Cefotaxime, CLI Clindamycin, NS Percentage not specified, CXM Cefuroxime, VAN Vancomycin, * It includes not only adults but also neonates and children, Classification as susceptible, resistant, or intermediate as reported in the study.
(DOCX)

# Acknowledgments

We thank Proma Paul from the London School of Hygiene and Tropical Medicine for encouraging the authors to do research on this topic and for her continuous guidance and support.

# Author Contributions

**Conceptualization:** Adoración Navarro-Torné, Daniel Curcio, Jennifer C. Moïsi.

**Data curation:** Adoración Navarro-Torné, Daniel Curcio.

**Formal analysis:** Adoración Navarro-Torné.

**Investigation:** Adoración Navarro-Torné, Daniel Curcio.

**Methodology:** Adoración Navarro-Torné.

**Project administration:** Adoración Navarro-Torné.

**Resources:** Adoración Navarro-Torné, Daniel Curcio.

**Software:** Adoración Navarro-Torné.

**Supervision:** Adoración Navarro-Torné, Jennifer C. Moïsi.

**Validation:** Adoración Navarro-Torné.

**Visualization:** Adoración Navarro-Torné.

**Writing – original draft:** Adoración Navarro-Torné.

**Writing – review & editing:** Adoración Navarro-Torné, Daniel Curcio, Jennifer C. Moïsi, Luis Jodar.

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
