## [Decision Letter · Decision Letter 0]

4 Aug 2021

PONE-D-21-20104

Burden of invasive group B Streptococcus disease in non-pregnant adults:  A systematic review and meta-analysis

PLOS ONE

Dear Dr. Curcio,

Thank you for submitting your manuscript to PLOS ONE. After careful consideration, we feel that it has merit but does not fully meet PLOS ONE’s publication criteria as it currently stands. Therefore, we invite you to submit a revised version of the manuscript that addresses the points raised during the review process.

The manuscript has been assessed by two reviewers. Their comments are available below.

One of the reviewers has raised a number of concerns about the clarity in the presentation of the work and the data, he/she recommends revisions to improve the clarity in presentation and writing and to provide a fuller outline of the methodology and main results.

Please carefully revise the manuscript to address all the points raised by the reviewers.

We look forward to receiving your revised manuscript.

Kind regards,

Jose Melo-Cristino, M.D., Ph.D.

Academic Editor

PLOS ONE

“The authors did not receive financial support for the conduct of this research apart from their salary as full-time employees of Pfizer.”

  a) Please provide an amended Funding Statement that declares *all* the funding or sources of support received during this specific study (whether external or internal to your organization) as detailed online in our guide for authors at http://journals.plos.org/plosone/s/submit-now. 

  b) Please state what role the funders took in the study.  If any authors received a salary from any of your funders, please state which authors and which funder. If the funders had no role, please state: "The funders had no role in study design, data collection and analysis, decision to publish, or preparation of the manuscript."

Please send your amended statements by return email; we will change the online submission form on your behalf.

3.  "Thank you for stating the following financial disclosure:

 “All authors are employees of Pfizer and may hold stock options. Pfizer supported the work.” 

Please respond by return e-mail so that we can amend your financial disclosure and competing interests on your behalf."

Reviewers' comments:

Reviewer's Responses to Questions

**Comments to the Author**

1. Is the manuscript technically sound, and do the data support the conclusions?

Reviewer #1: Yes

Reviewer #2: Yes

2. Has the statistical analysis been performed appropriately and rigorously? 

Reviewer #1: Yes

Reviewer #2: Yes

3. Have the authors made all data underlying the findings in their manuscript fully available?

Reviewer #1: No

Reviewer #2: Yes

4. Is the manuscript presented in an intelligible fashion and written in standard English?

Reviewer #1: Yes

Reviewer #2: Yes

5. Review Comments to the Author

Reviewer #1: In this systematic review the authors summarize the worldwide incidence and case fatality of invasive group B streptococcal disease in adults (excluding pregnant women). Database searches, quality appraisal and meta-analytical methods have been applied and reported appropriately, although I assume that a supplementary appendix was submitted which was not available for peer review (see comments #2 and #3 below)? The main limitations are duly acknowledged. This is a timely and well-presented systematic review and analysis. Timely because sufficient iGBS studies have accrued and, despite the heterogeneity between study estimates, there does appear to be an increase in incidence, albeit offset by a reduction in case fatality.

1. Abstract has a 95% CI written as "X-Y".

2. Quality appraisal - a summary of the QA needs to be given in the results and QA ratings provided in a supplementary file.

3. Data extraction - extracted data need to be provided in a supplementary file.

4. Results - a statement in the results that between-study heterogeneity was high (>60%) or very high (>90%) in most of the sub-groups would be helpful.

5. Results - meta-regression by time period - the p-value needs to be reported.

5. Discussion - where heterogeneity is discussed, I would be interested to know whether the authors could identify any possible reasons for major between-study differences in incidence and case-fatality (within a sub-group), e.g. Schrag (2000) vs. Tyrrell (2000) and others (where Schrag reported much higher incidence).

6. Discussion - I assume that a trend was not evident (statistically) because of the heterogeneity. As in, there is an apparent trend (doubling in incidence), but meta-regression won't detect it because of the between-study variation.

7. Figures - unfortunately Stata is not the best tool for creating Forest plots (I say that from experience)! Any additional tweaking that could be done, e.g. to report N without the +08 and to remove the sub-group subtotals where I2 and p are blank, would be welcome. I don't know if a newer version of Stata might bring improvements.

Reviewer #2: The authors conducted a systematic review and meta-analysis of GBS invasive disease in non-pregnant adults. The manuscript is of value to the field, revealing an overall increase in GBS infections as well as of antimicrobial resistance. The manuscript is well done, well written and fluent in all its parts. I have reviewed the paper from the perspective of a microbiologist, and I’m not confident I have the expertise to evaluate the statistical tools of meta-analyses; yet the methodologies used seem suitable for the major objectives proposed and the findings obtained are of value and well discussed. The conclusions are clearly reported and argued. The paper brings new insights into the dynamics of GBS population, contributing to a clearer picture of increasing incidence of non-neonatal group B streptococcal infections.

I have only one very minor comment: In the abstract (line 55), the 95% confidence interval for incidence rate of iGBS in ≥65 years old group is missing.

6. PLOS authors have the option to publish the peer review history of their article (what does this mean?). If published, this will include your full peer review and any attached files.

Reviewer #1: No

Reviewer #2: No

---

## [Author Response · Author response to Decision Letter 0]

14 Sep 2021

Response to reviewers

We would like to thank the reviewers for their comments that undoubtedly have contributed to improve the manuscript. 

Reviewer #1: In this systematic review the authors summarize the worldwide incidence and case fatality of invasive group B streptococcal disease in adults (excluding pregnant women). Database searches, quality appraisal and meta-analytical methods have been applied and reported appropriately, although I assume that a supplementary appendix was submitted which was not available for peer review (see comments #2 and #3 below)? The main limitations are duly acknowledged. This is a timely and well-presented systematic review and analysis. Timely because sufficient iGBS studies have accrued and, despite the heterogeneity between study estimates, there does appear to be an increase in incidence, albeit offset by a reduction in case fatality.

1. Abstract has a 95% CI written as "X-Y". Corrected. Page 2, line 55

2. Quality appraisal - a summary of the QA needs to be given in the results and QA ratings provided in a supplementary file. Corrected. QA results in page 13, lines 214-226. Table S4 in Supplementary information file

3. Data extraction - extracted data need to be provided in a supplementary file. Corrected. Supplementary information file provided

4. Results - a statement in the results that between-study heterogeneity was high (>60%) or very high (>90%) in most of the sub-groups would be helpful. Corrected. Statement provided in page 13, line 227

5. Results - meta-regression by time period - the p-value needs to be reported. Corrected. P values provided in page 14, line 244; page 15, line 255

5. Discussion - where heterogeneity is discussed, I would be interested to know whether the authors could identify any possible reasons for major between-study differences in incidence and case-fatality (within a sub-group), e.g. Schrag (2000) vs. Tyrrell (2000) and others (where Schrag reported much higher incidence). Discussed in page 18, lines 335-349

6. Discussion - I assume that a trend was not evident (statistically) because of the heterogeneity. As in, there is an apparent trend (doubling in incidence), but meta-regression won't detect it because of the between-study variation. Yes, thanks.

7. Figures - unfortunately Stata is not the best tool for creating Forest plots (I say that from experience)! Any additional tweaking that could be done, e.g. to report N without the +08 and to remove the sub-group subtotals where I2 and p are blank, would be welcome. I don't know if a newer version of Stata might bring improvements. Thank you for this remark. Figures look much better now!! Corrected. New corrected figures attached (+08 removed and full figures displayed I2 and p blank, deleted).

Reviewer #2: The authors conducted a systematic review and meta-analysis of GBS invasive disease in non-pregnant adults. The manuscript is of value to the field, revealing an overall increase in GBS infections as well as of antimicrobial resistance. The manuscript is well done, well written and fluent in all its parts. I have reviewed the paper from the perspective of a microbiologist, and I’m not confident I have the expertise to evaluate the statistical tools of meta-analyses; yet the methodologies used seem suitable for the major objectives proposed and the findings obtained are of value and well discussed. The conclusions are clearly reported and argued. The paper brings new insights into the dynamics of GBS population, contributing to a clearer picture of increasing incidence of non-neonatal group B streptococcal infections.

I have only one very minor comment: In the abstract (line 55), the 95% confidence interval for incidence rate of iGBS in ≥65 years old group is missing. Corrected. Page 2, line 55

---

## [Decision Letter · Decision Letter 1]

17 Sep 2021

Burden of invasive group B Streptococcus disease in non-pregnant adults:  A systematic review and meta-analysis

PONE-D-21-20104R1

Dear Dr. Curcio,

We’re pleased to inform you that your manuscript has been judged scientifically suitable for publication and will be formally accepted for publication once it meets all outstanding technical requirements.

Kind regards,

Jose Melo-Cristino, M.D., Ph.D.

Academic Editor

PLOS ONE

Additional Editor Comments (optional):

Reviewers' comments:

Reviewer's Responses to Questions

**Comments to the Author**

1. If the authors have adequately addressed your comments raised in a previous round of review and you feel that this manuscript is now acceptable for publication, you may indicate that here to bypass the “Comments to the Author” section, enter your conflict of interest statement in the “Confidential to Editor” section, and submit your "Accept" recommendation.

Reviewer #1: All comments have been addressed

2. Is the manuscript technically sound, and do the data support the conclusions?

Reviewer #1: Yes

3. Has the statistical analysis been performed appropriately and rigorously? 

Reviewer #1: Yes

4. Have the authors made all data underlying the findings in their manuscript fully available?

Reviewer #1: Yes

5. Is the manuscript presented in an intelligible fashion and written in standard English?

Reviewer #1: Yes

6. Review Comments to the Author

Reviewer #1: I am satisfied that the authors have address all of the comments that I made and would recommend this manuscript for acceptance. It is a well-conducted and very useful review and I believe that it merits publication.

7. PLOS authors have the option to publish the peer review history of their article (what does this mean?). If published, this will include your full peer review and any attached files.

Reviewer #1: No

---

## [Editor Report · Acceptance letter]

22 Sep 2021

PONE-D-21-20104R1 

Burden of invasive group B Streptococcus disease in non-pregnant adults:  A systematic review and meta-analysis 

Dear Dr. Curcio:

I'm pleased to inform you that your manuscript has been deemed suitable for publication in PLOS ONE. Congratulations! Your manuscript is now with our production department. 

Kind regards, 

on behalf of

Prof. Jose Melo-Cristino 

Academic Editor

PLOS ONE